# Swallowed topical steroid therapy for eosinophilic oesophagitis in children: practical, evidence-based guidance by the BSPGHAN Eosinophilic Oesophagitis Working Group

Joseph Chan [1], Diana M Flynn,[2] Morris Gordon [3] Raj Parmar,[4] Kerryn Moolenschot,[5] Lucy Jackman,[6] Ed Gaynor,[7] Jenny Epstein,[8] Amanda Cordell,[9] Hema Kannappan,[10] Mark Furman,[11] Julie Thompson,[12] Marco Gasparetto,[13] Marcus K H Auth[14]

For numbered affiliations see end of article.

**Correspondence to**
Professor Morris Gordon;
Mgordon@uclan.ac.uk

## ABSTRACT

**Objective** To develop evidence-based guidance for topical steroid use in paediatric eosinophilic oesophagitis (pEoE) in the UK for both induction and maintenance treatment.

**Methods** A systematic literature review using Cochrane guidance was carried out by the British Society of Paediatric Gastroenterology, Hepatology and Nutrition (BSPGHAN) Eosinophilic Oesophagitis (EoE) Working Group (WG) and research leads to determine the evidence base for preparation, dosing and duration of use of swallowed topical steroid (STS) formulations in EoE. Seven themes relating to pEoE were reviewed by the WG, alongside the Cochrane review this formed the evidence base for consensus recommendations for pEoE in the UK. We provide an overview of practical considerations including treatment regimen and dosing. Oral viscous budesonide (OVB) and, if agreed by local regulatory committees, orodispersible budesonide (budesonide 1 mg tablets) were selected for ease of use and with most improvement in histology. A practical 'how to prepare and use' OVB appendix is included. Side effects identified included candidiasis and adrenal gland suppression. The use of oral systemic steroids in strictures is discussed briefly.

**Results** 2638 citations were identified and 18 randomised controlled trials were included. Evidence exists for the use of STS for induction and maintenance therapy in EoE, especially regarding histological improvement. Using the Appraisal of Guidelines, Research and Evaluation criteria, dosing of steroids by age (0.5 mg two times per day <10 years and 1 mg two times per day ≥10 years) for induction of at least 3 months was suggested based on evidence and practical consideration. Once histological remission is achieved, maintenance dosing of steroids appears to reduce the frequency and severity of relapse, as such a maintenance weaning regimen is proposed.

**Conclusion** A practical, evidence-based flow chart and guidance recommendations with consensus from the EoE WG and education and research representatives of BSPGHAN were developed with detailed practical considerations for use in the UK.

---

## WHAT IS ALREADY KNOWN ON THIS TOPIC

⇒ Therapeutic options for paediatric eosinophilic oesophagitis are limited. Swallowed topical steroids are the most successful treatment option in adults and children.

⇒ Unlike in adults, currently there is no licensed formulation for a topical steroid in paediatric eosinophilic oesophagitis leaving prescribers with a variety of local dispensing policies.

⇒ These therapies have evidence of benefit for paediatric patients of all ages but exact data on efficacy and safety have been missing in previous guidelines.

---

## INTRODUCTION

Eosinophilic oesophagitis (EoE) is a chronic, immune or antigen-mediated oesophageal disease characterised by symptoms of oesophageal dysfunction and eosinophil predominant mucosal inflammation[1] first discovered in the early 1990s.[2 3] The rising prevalence is estimated at 15/100 000 before 2007 and 63/100 000 since 2017.[4] It is more common in males and is associated with atopic diseases. Current incidence estimates range from 5 to 10/100 000,[5] similar to paediatric inflammatory bowel disease in the UK.[6] There has been a need to produce guidance for healthcare professionals and patients, but this has mostly been adult focused and not always translatable to paediatric medicine. In the UK, a joint consensus guideline by the British Society of Gastroenterology and the British Society of Paediatric Gastroenterology, Hepatology and Nutrition (BSPGHAN) has attempted to redress this balance and pave the way for a joined-up EoE approach.[7]

Swallowed topical steroids (STS) have become common practice in recent years for

**WHAT THIS STUDY ADDS**

⇒ Swallowed topical steroids probably provide improved remission rates for clinical symptoms and histology in children not responding to proton pump inhibitor medication.

⇒ In children <10 years old, a standard total daily dose of 1 mg/day can be safely used. In children ≥10 years old, a standard total daily dose of 2 mg/day should be used.

⇒ Expert consensus would favour a minimum of a 3-month induction period, followed by repeat endoscopy with biopsies.

⇒ Maintenance therapy should be considered for a minimum of 1–2 years. Usually, maintenance dosing is started at half the daily induction dose.

⇒ In children, histological findings often correlate very poorly with symptoms.

**HOW THIS STUDY MIGHT AFFECT RESEARCH, PRACTICE OR POLICY**

⇒ These recommendations and flow chart will help standardise medical care of children with eosinophilic oesophagitis treated with swallowed topical steroids both nationally and internationally.

⇒ Implementing a robust scheme for induction and maintenance with mandatory surveillance endoscopies will allow local benchmarking and prospective outcome measures.

⇒ This study will close a gap of clinical research and provide a bridge for future studies on best treatment options and prognosis for children with eosinophilic oesophagitis.

induction and maintenance treatment of EoE; however, practical barriers to treatment exist for the paediatric EoE (pEoE) cohort. The diagnosis is normally made in tertiary paediatric gastroenterology settings due to access to endoscopy and paediatric anaesthetists for general anaesthesia, these settings are less accessible for families living outside of major cities. Additionally, the newest orodispersible STS is unlicensed in patients less than 18 years old and is often unavailable in primary and secondary care. Tertiary centres are obliged to continue prescriptions which contributes to increased cost burden to families making long journeys to collect the medication or on hospital pharmacies who fund delivery of the medication to the family. Another barrier for pEoE is obtaining STS that are child-friendly and acceptable to the individual and family. Poor compliance is associated with medications that are difficult to administer due to product bitterness, time taken to prepare the medication and technical challenges involved in coordinating the swallow in younger age groups.

In the absence of a licensed paediatric formulation, choice of different STS and inequality of access to tertiary healthcare settings, UK practice varies and is not standardised. The BSPGHAN EoE Working Group (WG) has taken the opportunity to collaborate with BSPGHAN research leads and used Cochrane evidence to address some of the issues faced by children with EoE and provide healthcare professionals with practical, evidence-based guidance and a flow chart to support clinical care in healthcare settings.

## METHODS

The BSPGHAN EoE WG met virtually by teleconference to discuss issues relevant to pEoE and what aspects guidance should focus on. There were four meetings between December 2021 and December 2022. The authors performed a systematic literature review on CENTRAL, MEDLINE, Embase, ClinicalTrials.gov and WHO ICTRP on 3 March 2023, with no prior date limits, using Cochrane guidance.[8] Search criteria were randomised controlled trials (RCTs) comparing any medical intervention or food elimination diet for the treatment of EoE. Dichotomous analysis was undertaken to assess response, either alone or in combination to any other intervention (including placebo). 2638 citations were identified and 18 RCTs pertaining to steroid use were included. This technical review and GRADE (Grading of Recommendations, Assessment, Development, and Evaluations) evidence included both adult and paediatric studies and informed the evidence base for the WG recommendations.

The authors and WG reviewed the following seven themes in paediatric patients (aged <18 years) diagnosed with EoE:

1. What are the indications for STS?
2. Which STS preparations are best and how should they be made up?
3. Should dosing be based on age or height?
4. What dose and duration should induction therapy be?
5. What dose and duration should maintenance therapy be?
6. How and when should STS be reduced or stopped?
7. When should endoscopy be repeated?

Recommendations related to each theme were proposed and the WG voted anonymously on each recommendation. A 9-point scale was used, with 9 representing strong agreement and 1 representing strong disagreement. Consensus was reached if more than 80% of eligible voting members voted 6, 7, 8 or 9. All WG members are professionals with expertise in EoE.

### Patient and public involvement

Two members of the WG are affiliated with patient support groups; one is the chief executive officer of the EOS Network, an Eosinophilic Disease Charity that supports patients with EoE and their families; the other member is affiliated with Guts UK, a charity funding research and supporting people living with gastrointestinal disease. Their involvement has been from the outset of development of this project with contribution and comment on the flow chart, statements and recommendations. They have been integral in considering how the information can be disseminated to the wider EoE community.

### RESULTS

All of the statements made by the BSPGHAN EoE WG (table 1) for each pEoE theme reached consensus. Voting breakdowns are provided (online supplemental file 1).

**Table 1** Recommendations from the BSPGHAN EoE WG with voting consensus percentage

| Theme | The BSPGHAN EoE WG recommends that: | Voting consensus (%) |
|---|---|---|
| 1 | **STS are indicated in:** | |
| | Induction therapy | 100 |
| | Maintenance therapy | 100 |
| | First-line treatment of EoE | 92 |
| | Combination therapy with PPI | 83 |
| | Second-line treatment after unsuccessful responses to dietary therapy and/or PPI | 100 |
| | Oesophageal stricture treatment in isolation or as an adjunct to oesophageal balloon dilatation | 92 |
| 2 | The type of STS preparation offered should take into consideration the child's age, palatability, chance of adherence, comorbidities and family support | 100 |
| | If a child is assessed to be able to tolerate and coordinate orodispersible budesonide (tablet) preparation and it is locally available, then this preparation should be first choice. | 92 |
| 3 | Age should be used to determine STS dosing unless a child significantly deviates from their growth centiles, in which case height should be accounted for. | 100 |
| 4 | Twice daily dosing is considered for induction therapy. | 100 |
| | For children <10 years old, a dose of 0.5 mg two times per day should be used (1 mg/day), for children ≥10 years old, a dose of 1 mg two times per day should be used (2 mg/day). | 100 |
| | Induction therapy should be usually given for a minimum of 3 months. | 100 |
| 5 | After histological remission has been achieved, maintenance therapy should be considered for a minimum of 1–2 years. | 100 |
| | For children <10 years old, a dose of 0.5 mg/day should be used, for children ≥10 years old, a dose of 1 mg/day should be used. | 100 |
| 6 | After confirmed histological remission and 3-month induction therapy, clinicians should consider halving the STS dose. | 100 |
| | During maintenance therapy, dose weaning should be considered every 6–12 months. | 92 |
| | Oral or oesophageal candidiasis does not usually require stopping the STS; antifungal treatment should be added alongside the STS. | 83 |
| 7 | **Endoscopy** | |
| | Should be repeated during the induction period to ensure histological response to STS and allow weaning of the medication. | 92 |
| | Should be repeated if there is worsening of symptoms/oesophageal dysfunction. | 100 |
| | Surveillance should be considered at 1–2 yearly intervals or if considering stopping treatment or following cessation of therapy if clinically indicated. | 100 |

BSPGHAN EO WG, British Society of Paediatric Gastroenterology, Hepatology and Nutrition Eosinophilic Oesophagitis Working Group; PPI, proton pump inhibitor; STS, swallowed topical steroid.

## Theme 1: what are the indications for STS?

There is strong evidence for the use of STS in EoE. STS have very good remission rates, 71% symptomatically and 59% histologically in children not responding to proton pump inhibitor (PPI) medication.[8 9] There is emerging evidence showing that combination therapy of STS with PPI may lead to even better outcomes.[8] Currently, there are no data in pEoE showing that the use of STS will prevent oesophageal strictures or that strictures can be treated in isolation with STS. However, in practice, medical treatment is often trialled before therapeutic balloon dilatation. The main evidence supporting steroid use in moderate to severe oesophageal strictures is by using systemic steroids. A multicentre retrospective cohort study found that with systemic steroids 20/20 patients showed clinical improvement, 15/20 became asymptomatic and 19/20 had stricture resolution at endoscopy.[10]

## Theme 2: which STS preparations are best and how should they be made up?

There are three main STS preparations: (a) oral viscous budesonide ('budesonide slurry') formed from budesonide respules combined with a viscous binding agent, (b) orodispersible budesonide (tablets) which melt on the tongue before being swallowed and (c) inhaled fluticasone sprayed into the buccal cavity and swallowed. There are currently no head-to-head trials on different preparations or on the ideal viscous binding agent to use to form the slurry. Expert opinion would advocate using an age

appropriate preparation which is generally a budesonide slurry in younger children and orodispersible tablets in older children.

### Theme 3: should dosing be based on age or height?

Research studies looking at histological remission rates and dosing in children with EoE have used either age ranges (<10 or ≥10 years) or heights (<1.5 or ≥1.5 m) to define cut-offs for medication dosing.[11] Expert opinion would advocate using age for simplicity. If a child deviates from their expected growth centiles then consideration should be given to using height instead of age to determine STS dosing.

### Theme 4: what dose and duration should induction therapy be?

There is a lack of evidence surrounding the duration and dosing regimen of induction therapy. Expert consensus would favour a minimum of a 3-month induction period and twice daily dosing to promote increased contact time between the medication and oesophageal mucosa (most clinical studies have been conducted with twitch twice daily dosing).

In adults, data from randomised placebo-controlled trials show that in a dosage of 1 mg two times per day, orodispersible budesonide achieved remission after 12 weeks in 85% of patients.[12] Viscous budesonide formulations using sucralose or other substrates (evaluated in a prospective randomised placebo-controlled trial only) may be equally, less or more effective.

In children <10 years old, a standard total daily dose of 1 mg/day in divided doses can be safely used. In children ≥10 years old, a standard total daily dose of 2 mg/day in divided doses should be used. In select cases, doses of up to 2.8 mg/day (<10 years old) and 4 mg/day (≥10 years old) have been used.[13] Clinician judgement should be used to balance up the optimal dosing frequency versus the chance of adherence in particular cases.

### Theme 5: what dose and duration should maintenance therapy be?

Once histological remission has been accomplished, use of maintenance dosing of steroids appears to reduce the frequency and severity of relapse.[9 14] There is currently limited evidence regarding how long maintenance therapy should be continued for. In adults, data from randomised placebo-controlled trials show that a dosage of orodispersible budesonide of 0.5 or 1 mg two times per day achieved persistent remission after 48 weeks in 73.5% and 75%, respectively.[14] The viscous formulations using sucralose or other substrates (evaluated in a prospective randomised placebo-controlled trial only) may be equally, better or less effective.

Once histological remission has been achieved during the induction phase, maintenance therapy should be considered for a minimum of 1–2 years. Usually, maintenance dosing is started at half the daily induction dose. In the majority of published clinical studies, STS are given two times per day (18 studies), and only in three studies once daily (evening) and one study four times per day.[9] Outside of clinical studies, in reality, as advised by our corresponding patient support organisations, patients and families sometimes struggle with the logistics of more frequent (more than once daily) dosing, with the morning dose often missed.[15] Therefore, we recommend that consideration should be given and discussion take place to whether once or twice daily dosing is likely to result in improved adherence to longer term therapy and achievement of the desired effect. There are no data suggesting that twice daily dosing determines an increased risk, and no data indicating that once daily dosing has the same treatment effect. From a practical perspective, principles of dental hygiene and exposure of the oesophagus to the drug for over 30 min with no food or drink allowed determine which logistics are feasible for children and families on a daily basis.

### Theme 6: how and when should STS be reduced or stopped?

Histological remission is defined as <15 eosinophils per high power field or 0.3 mm$^2$ in any oesophageal biopsy examined (proximal, mid and distal oesophagus). Systemic side effects of topical steroids are considered rare during the long-term treatment of patients with EoE in those naïve to STS and without pre-existing adrenal conditions.[16] However, patients with EoE often have associated atopic conditions, and careful consideration should be given to the total daily steroid dose and steroid burden that these patients receive especially during seasonal exacerbations of their atopic disease. Consideration should be given to maintaining EoE on the lowest practical dose of STS to maintain histological remission.

In a systematic review on adrenal insufficiency in children with EoE, adrenal testing was abnormal in 15.8%; however, adrenal insufficiency between topical steroids and placebo was not statistically different over 2–12 weeks of treatment.[17] In observational studies, the risk of adrenal insufficiency increased from 0–10% up to 30–66% in patients with multiple steroid treatments (eg, for asthma or rhinitis).[17] In a multicentre study from the USA, 5% of children on STS were considered to show signs of adrenal gland insufficiency while on STS for longer than 6 months.[16]

Children with symptoms suggestive of adrenal insufficiency or risk factors for developing the condition (eg, patients receiving multiple steroids or on twice daily induction regimens for more than 6 months) should be evaluated as per local protocols. The short synacthen test (low-dose injected adrenocorticotropic hormone) is the preferred investigation in most centres.[17]

Continued monitoring of growth, bone mineral density and adrenal suppression is recommended in children and adolescents.[7] Subtle decrease in linear growth was observed in a retrospective multicentre study in children with EoE receiving topical steroid treatment, and sex differences were observed affecting girls treated with combined elemental diet and topical steroids.[18] Children

| |
|---|
| 1. Corticosteroids lead to a large histological improvement (63% higher), measured dichotomously (RR 11.94, 95% CI 6.56 to 21.75; 12 studies, 978 participants; NNTB = 3; high certainty), and may lead to histological improvement, measured continuously (SMD 1.42, 95% CI 1.02 to 1.82; 5 studies, 449 participants; low certainty). |
| 2. Corticosteroids may lead to slightly better clinical improvement (20% higher), measured dichotomously (risk ratio (RR) 1.74, 95% CI 1.08 to 2.80; 6 studies, 583 participants; number needed to treat for an additional beneficial outcome (NNTB) = 4; low certainty), and may lead to slightly better clinical improvement, measured continuously (standard mean difference (SMD) 0.51, 95% CI 0.17 to 0.85; 5 studies, 475 participants; low certainty). |
| 3. Corticosteroids may lead to little to no endoscopic improvement, measured dichotomously (RR 2.60, 95% CI 0.82 to 8.19; 5 studies, 596 participants; low certainty), and may lead to endoscopic improvement, measured continuously (SMD 1.33, 95% CI 0.59 to 2.08; 5 studies, 596 participants; low certainty). |
| 4. Corticosteroids may lead to slightly fewer withdrawals due to adverse events (RR 0.64, 95% CI 0.43 to 0.96; 14 studies, 1032 participants; low certainty) and patients probably experience a similar number of both serious side effects and side effects in total, compared to placebo. There may be no difference between corticosteroids and placebo in the improvement of quality of life. |

**Figure 1** Key evidence from the Cochrane review. The Cochrane review provides analysis of 52 statements on the use of swallowed topical steroid (STS) for the endpoints; successful reduction of eosinophils, resolution of symptoms (dichotomous), mean drop in eosinophils and mean reduction in symptom score (continuous).[8] The key findings are from 41 randomised controlled trials (RCTs) with 3253 participants, of which 14 studies compared corticosteroids to placebo for induction of remission in children alone or children and adults.

with nutritional deficits (eg, secondary to behavioural or dietetic restrictions) should be carefully monitored for adequate growth (eg, centile, nutritional assessment, biochemically) and bone health (eg, DEXA (dual energy x-ray absorptiometry) scan).[19]

### Theme 7: when should endoscopy be repeated?
There is no clear paediatric guidance on when an endoscopy should be repeated following diagnosis. In children, histological findings often correlate very poorly with symptoms.[9 20] Endoscopy and biopsy should be used to ensure that STS treatment strategies are effective, enable management changes and investigate worrying symptoms.

### Flow chart
By assimilating the key evidence from the literature and Cochrane review (figure 1), recommendations (table 1) and expert consensus, a practical easy-to-use flow chart (figure 2) was designed to help support healthcare professionals involved in the management of pEoE.

### DISCUSSION
The incidence, prevalence, recognition and management of EoE has rapidly changed over the past 30 years and will continue to evolve as our understanding and therapeutic options change. It is essential that guidance and recommendations are standardised to ensure patients are not disadvantaged by variations in access to healthcare professionals, specific professional knowledge or medications. Using the latest Cochrane systematic review and through collaboration between a national EoE expert group, BSPGHAN education and research leads and patient support groups, we recommend

guidance that addresses common themes in pEoE in the UK. We have produced a quick reference flow chart that can be used at the bedside to help guide professionals involved in the management of pEoE. The flow chart was orally presented at the 2023 BSPGHAN Annual Meeting attended by 130 professionals for wider society member review and discussion. Specific comments were requested using a QR code to ensure stakeholder engagement before wider dissemination. The flow chart was found to be useful by all responders with respondents keen to use it in their practice. Where evidence exists, this has been used to strengthen the recommendations and flow chart and where lacking, expert consensus has been followed.

### Efficacy, palatability and adherence
Twice daily dosing theoretically allows for a minimum of a daily anti-inflammatory dose especially if one dose is missed. The WG were advised that for adherence to maintenance therapy once daily dosing is more acceptable.[15] Individual circumstances should be discussed and tailored according to lifestyle and activities. As there are currently no licensed preparations of STS in children, a variety of carrier substances for budesonide STS preparations to coat the oesophagus have been used for administration and palatability (online supplemental appendix 1).[21–26] This is heavily reliant on local expertise and there are no head-to-head trials to advocate use of one over another. Orodispersible budesonide (tablets) are licensed in adults and are increasingly being used off-label. Prospective paediatric studies for STS have been conducted and first results have been submitted for peer-reviewed presentation,[27] indicating good prospects for suspensions and tablets, subject to pharmaceutical licensing, for future application in children.

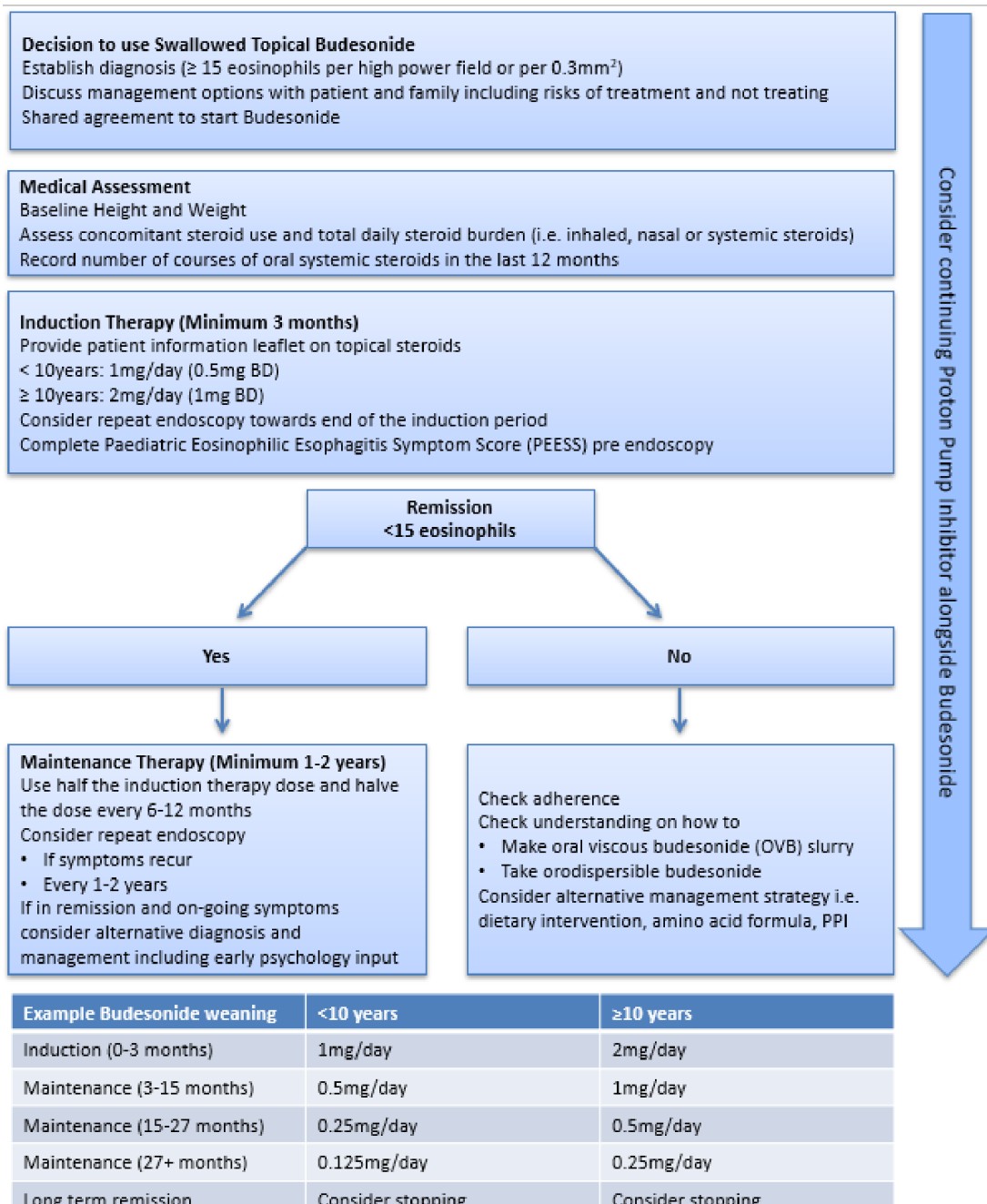

**Figure 2** Swallowed topical budesonide flow chart (quick clinical reference guide). BD, two times per day; PPI, proton pump inhibitor.

## Side effects

Children with EoE have a 70% association with atopic comorbidity, steroids are often prescribed by other physicians (dermatology, respiratory, general paediatrics) which potentially increases the cumulative steroid burden. The largest multicentre study reported that 5% of children showed signs of adrenal gland insufficiency while on STS for EoE for longer than 6 months.[16] In study protocols, baseline cortisol levels, short synacthen testing and follow-up while on maintenance treatment have been applied; however, there is controversy of the standardisation and interpretation of test results, we recommend

performing these as clinically indicated and according to local endocrinology advice. Other monitoring should include growth parameters and bone mineral densities. *Candida* infection is rare and can be treated simultaneously with topical antifungal or oral medication.[7]

## Endoscopy, prognosis, withdrawal and escalation

In the absence of validated clinical activity scores or non-invasive biomarkers, the group agree that endoscopic reassessment with histology should be used after any significant treatment start or modification. As STS are effective in more than 80% of children when taken according to

protocols and usually well tolerated, prognosis is very good. Withdrawal is indicated in non-responders, disease progression under therapeutic dosage of STS or clinically relevant adrenal gland insufficiency. We found limited evidence for escalation of treatment with regard to dosage of STS.[18 20]

## Limitations

There is no evidence for the ideal carrier substance used to generate the optimal formulation of STS. There is a lack of studies to compare potential effects of once or twice daily administration of a defined daily dose of STS on efficacy and long-term outcomes. STS have been considered mainly in isolation and combination therapy has not been investigated as they have been discussed elsewhere.[8 18 20 28]

## Implications for clinicians and policymakers

This project is a starting point for pEoE in the UK and beyond; further work should be considered with other professionals in paediatric practice to develop a collaborative multisystem approach to pEoE care including allergy and transition to adult settings. This would ensure continued evolution of the clinical reference flow chart when better access to medication occurs; merging paediatric and adult practice should be the gold standard to aim for.

## Unanswered questions and future research

Increasing the pEoE knowledge base is vital to combat the increasing disease burden. Differentiating if the increased pEoE prevalence is due to earlier recognition or is suggestive of a more severe phenotype may help risk stratify patients and impact ongoing surveillance and transition decisions. While it is important to research new treatments for EoE and improve access to these in paediatrics, it will be crucial to know whether the stimulation and recruitment of eosinophils into the oesophagus can be prevented. More trials are needed on whether the type, viscosity and volume of the carrier solution affect the efficacy of topical budesonide. Using this guidance as a baseline will allow future research into best practice and prognosis of pEoE.

## CONCLUSION

Evidence-based guidance and a flow chart with consensus from the EoE WG and education and research representatives of BSPGHAN were developed with detailed practical considerations for use in the UK and beyond.

## Author affiliations

[1]Paediatric Gastroenterology, University Hospital of Wales, Cardiff, UK
[2]Paediatric Gastroenterology, Royal Hospital for Children, Glasgow, UK
[3]University of Central Lancashire, Preston, UK
[4]Paediatric Gastroenterology, Alder Hey Children's Hospital, Liverpool, UK
[5]Clinical Lead Paediatric Dietitian, St George's Hospital, London, UK
[6]Specialist Paediatric Dietitian, Great Ormond Street Hospital for Children, London, UK
[7]Paediatric Gastroenterology/Mucosal Immunology, Great Ormond Street Hospital for Children, London, UK
[8]Paediatric Gastroenterology, Chelsea and Westminster Hospital, London, UK
[9]Eosinophilic Family Coalition, London, UK
[10]General Paediatrics, University Hospitals Coventry and Warwickshire NHS Trust, Coventry, UK
[11]Paediatric Gastroenterology, Royal Free Hospital, London, UK
[12]Guts UK, London, UK
[13]Paediatric Gastroenterology, Norfolk and Norwich University Hospital, Norwich, UK
[14]Gastroenterology, Hepatology and Nutrition, Alder Hey Children's NHS Foundation Trust, Liverpool, UK

**Collaborators** BSPGHAN Eosinophilic Oesophagitis Working Group: Drs Marcus Karlheinz Auth, Joseph Chan, Diana M Flynn—details as above; Amanda Cordell, EOS Network—Eosinophilic Diseases Charity; Dr Jenny Epstein, Chelsea and Westminster Hospital, London; Dr Mark Furman, Centre for Paediatric Gastroenterology, Royal Free Hospital, London; Dr Ed Gaynor, Great Ormond Street Hospital, London; Lucy Jackman, Great Ormond Street Hospital, London; Dr Hema Kannappan, University Hospital Coventry and Warwickshire; Kerryn Moolenschot, St George's Hospital, London; Dr Raj Parmar, Alder Hey Children's Hospital; Julie Thompson, Guts UK, Guts UK Charity.

**Contributors** JC and DMF had the idea for this project on behalf of the BSPGHAN EoE Working Group. The initial paper search and outline of the article were developed by them. JC took the lead on writing the paper. DMF contributed to the writing of the paper and development of the steroid algorithm. MG performed the detailed Cochrane review whereby corroborating evidence was obtained and contributed to structure and writing of the paper. MG contributed to the writing of the paper, and advice on grading of statements and recommendations. MKHA took senior overview of the development of the paper and algorithm as chair of the BSPGHAN EoE Working Group, and contributed major to writing and editing of the paper. All members of the BSPGHAN EoE Working Group contributed to review of the steroid algorithm, voting and corroboration of statements. The guarantor for the accuracy and clinical governance of the manuscript is MKHA as chair of the EoE Working Group and member of ESPGHAN EGID Working Group.

**Funding** The authors have not declared a specific grant for this research from any funding agency in the public, commercial or not-for-profit sectors.

**Competing interests** MKHA has received research grants and honorariums from BSPGHAN, Guts UK, Dr Falk Pharma, Mead Johnson, Abbott and Nutricia; AC has received grant funding from AZ, Bristol Myers Squibb, AVIR, Allakos, Dr Falk Pharma and Sanofi/Regeneron, and advisory panel honorarium was received from AZ, Sanofi/Regeneron, AVIR and Dr Falk Pharma; Guts UK Charity receives funding for the Falk Awards from Dr Falk Pharma.

**Patient and public involvement** Patients and/or the public were involved in the design, or conduct, or reporting, or dissemination plans of this research. Refer to the Methods section for further details.

**Patient consent for publication** Not applicable.

**Provenance and peer review** Not commissioned; externally peer reviewed.

**Data availability statement** All data relevant to the study are included in the article or uploaded as supplementary information.

**ORCID iDs**
Joseph Chan http://orcid.org/0000-0002-7281-9619
Morris Gordon http://orcid.org/0000-0002-1216-5158

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
