## [Reviewer comments · BMJ Paediatrics Open]

ARTICLE DETAILS

TITLE (PROVISIONAL)	Swallowed topical steroid therapy for eosinophilic oesophagitis in children: Practical, evidence-based guidance by the BSPGHAN eosinophilic oesophagitis working group
AUTHORS	Chan, Joseph Flynn, Diana Gordon, Morris Parmar, Raj Moolenschot, Kerry Jackman, Lucy Gaynor, Ed Epstein, Jenny Cordell, Amanda Kannappan, Hema Furman, Mark Thompson, Julie Gasparetto, Marco Auth, Marcus

VERSION 1 – REVIEW

REVIEWER	Hamish Philpott United Kingdom of Great Britain and Northern Ireland
REVIEW RETURNED	21-Jan-2024

GENERAL COMMENTS	This large body of work follows a standard format, aiming for a structured literature review and a working group then issued guidelines regarding EoE management with topical corticosteroids. I note both adult and paediatric studies were reviewed. I also note the studies that are specifically referenced to support the guidelines or more specifically the 7 themes (methods section) I have no objection to the methods or themes - these seem good, as is the introductory material. I do however think the rationale for the use of both induction and maintenance dosing and dose interval needs to be clarified. Basically, randomised placebo controlled data for maintenance treatment needs to be considered. The oral dispersible formulation Jorveza has been studied out to >48 weeks, utilising 0.5mg po BD and 1mg Po BD in adults. This demonstrates >80% efficacy. This data should be specifically included, and a comment that an ad-hoc formulation made up using sucralose (splenda) may well be equally effective, albeit less effective. **I do challenge the authors to back up the suggestion that daily as opposed to twice daily treatment is as effective? please support this and add a comment as to the likely decrement in efficacy based on the available evidence.
---

	**I suggest commenting that in patients with multiple steroid treatments (e.g. for asthma or rhinitis as well) the adrenal suppression may be a greater risk (reference Philpott Dellon et al systematic review in APT about 2018 or 2019) **I suggest commenting as to if any significant increased risk with twice daily dosing **I note there is a very weak statement about tests for adrenal suppression. In my view screening tests for adrenal suppression are not justified. I would suggest revisiting this comment and adding which test is needed and for whom
--	--

REVIEWER	DMeyer's Childrens' Hospital, Rambam Health Care Campus Gastroenterology Haifa Israel. Ron Shaoul
REVIEW RETURNED	22-Feb-2024

GENERAL COMMENTS	The paper is important and well written. I miss information on the effect of Swallowed topical steroid therapy on growth and the following paper by Jensen et al should be quoted. J Pediatr Gastroenterol Nutr . 2019 Jan;68(1):50-55.
--

VERSION 1 – AUTHOR RESPONSE

Dear Dr Chhooara, Dr Raman, and BMJ Paediatrics Open editorial board,
Dear reviewers Dr Philpott and Dr Shaoul,

bmjpo-2023-002467: Swallowed topical steroid therapy for eosinophilic oesophagitis in children: Practical, evidence-based guidance by the BSPGHAN eosinophilic oesophagitis working group by J Chan, et al

Thank you very much for inviting us to resubmit our manuscript with minor revisions. We are delighted that our work has been given consideration and interest from BMJ Paediatrics Open and believe that the prospect of publishing it with open access will provide a valuable asset for many readers and professional societies. We are grateful to the editors and reviewers for their excellent comments and suggestions, which have all been addressed and resolved, and have resulted in further improvement of the quality of the manuscript. Please find below our step by step response.

Editor(s)' Comments, Associate Editor:

Abstract Results. The last 5 sentences should be in Methods not Results.

We have moved these sentences from Results to Methods.

Key Messages avoid abbreviations.

We have changed all abbreviations into full wording.

What this study adds -please be more specific. You make excellent recommendations in the section on Themes -put some of these in instead of existing text.

Thank you for the excellent suggestion. We have removed the previous statements and replaced them with five key messages taken as extracts from our recommendations.

Reviewer 1, Hamish Philpott:

I do however think the rationale for the use of both induction and maintenance dosing and dose interval needs to be clarified. Basically, randomised placebo controlled data for maintenance treatment needs to be considered. The oral dispersible formulation Jorveza has been studied out to >48 weeks, utilising 0.5mg po BD and 1mg Po BD in adults. This demonstrates >80% efficacy. This data should be specifically included, and a comment that an ad- hoc formulation made up using sucralose (splenda) may well be equally effective, albeit less effective.

Thank you for the excellent advice. We have rewritten the Theme4 and 5 statements, quoting specifically the reference above and published evidence on induction and maintenance treatment.

I do challenge the authors to back up the suggestion that daily as opposed to twice daily treatment is as

effective? please support this and add a comment as to the likely decrement in efficacy based on the available evidence.

Thank you for the excellent advice. We have rewritten the Theme 4 and 5 statements, pointing out that the vast majority of studies for topical steroids in EoE were conducted with twice daily treatment, and have added a comment in two sections that if logistics of drug administration require a once daily approach, treatment may be less effective.

I suggest commenting that in patients with multiple steroid treatments (e.g. for asthma or rhinitis as well) the adrenal suppression may be a greater risk (reference Philpott Dellon et al systematic review in APT about 2018 or 2019)

Thank you for the excellent advice and the reference. We have amended the section on adrenal insufficiency in Theme 6, and have included the above mentioned, valuable reference.

I suggest commenting as to if any significant increased risk with twice daily dosing

Thank you for the advice. We have rewritten the Theme 5 statement, pointing out that the vast majority of studies for topical steroids in EoE were conducted with twice daily treatment, and have added a comment about the published safety of twice daily dosing.

I note there is a very weak statement about tests for adrenal suppression. In my view screening tests for adrenal suppression are not justified. I would suggest revisiting this comment and adding which test is needed and for whom

Thank you for the excellent advice and the reference. We have amended in Theme 6 the section on adrenal insufficiency, have included the above mentioned, valuable reference, and have added advice on which tests are needed for which cohort of patients (especially those at risk for potential side-effects of treatment).

Reviewer 2, Dr. Ron Shaoul:

I miss information on the effect of Swallowed topical steroid therapy on growth and the following paper by Jensen et al should be quoted. J Pediatr Gastroenterol Nutr . 2019 Jan;68(1):50-55.

Thank you very much for the excellent advice and the reference. We have amended the section on growth, have included the above mentioned, valuable reference, and have added advice on monitoring of growth especially in children at risk for growth impairment.

We thank the reviewers for their expertise and valuable time taken and hope that they are satisfied with our thorough amendments. With great anticipation we look forward to the decision of the editorial board of BMJ Paediatrics Open, with all suggestions being fully addressed.

We are willing to share the data from this study for open access.

Yours sincerely

VERSION 2 – REVIEW

REVIEWER	Hamish Philpott United Kingdom of Great Britain and Northern Ireland
REVIEW RETURNED	12-May-2024
GENERAL COMMENTS	All seems in order

VERSION 2 – AUTHOR RESPONSE

None